# Improving the Management of Endometrial Cancer Patients through the Use of Liquid Biopsy Analyses: A Case Report

**DOI:** 10.3390/ijms23158539

**Published:** 2022-08-01

**Authors:** Carlos Casas-Arozamena, Alexandra Cortegoso, Raquel Piñeiro-Perez, Alicia Abalo, Efigenia Arias, Victoria Sampayo, Ana Vilar, Marta Bouso, Eva Diaz, Gema Moreno-Bueno, Rafael López-López, Laura Muinelo-Romay, Miguel Abal, Juan Cueva

**Affiliations:** 1Translational Medical Oncology Group (Oncomet), Health Research Institute of Santiago de Compostela (IDIS), University Hospital of Santiago de Compostela (SERGAS), Trav. Choupana s/n, 15706 Santiago de Compostela, Spain; carlos.casas_95@hotmail.es (C.C.-A.); raquel.pineiro.perez@rai.usc.es (R.P.-P.); alicia.abalo.pineiro@sergas.es (A.A.); rafa.lopez.lopez@gmail.com (R.L.-L.); lmuirom@gmail.com (L.M.-R.); 2Department of Medicine, University of Santiago de Compostela (USC), Praza do Obradoiro, 0, 15705 Santiago de Compostela, Spain; 3Department of Medical Oncology, University Clinical Hospital of Santiago de Compostela, University of Santiago de Compostela (USC), 15706 Santiago de Compostela, Spain; ale.cm.acm@gmail.com; 4Department of Gynecology, University Clinical Hospital of Santiago de Compostela (SERGAS), Trav. Choupana s/n, 15706 Santiago de Compostela, Spain; efi.arias@yahoo.com (E.A.); vitosampayo@hotmail.com (V.S.); ana.vilar.lagares@sergas.es (A.V.); 5Department of Pathology, University Clinical Hospital of Santiago de Compostela (SERGAS), Trav. Choupana s/n, 15706 Santiago de Compostela, Spain; marta.bouso.montero@sergas.es; 6Foundation MD Anderson International, C/Gómez Hemans 2, 28033 Madrid, Spain; eva.diaz@fundacionmdanderson.es (E.D.); gema.moreno@fundacionmdanderson.es (G.M.-B.); 7Biochemistry Department, Medicine Faculty, Universidad Autónoma de Madrid (UAM), C/Arturo Dupurier 4, 28029 Madrid, Spain; 8Centro de Investigación Biomédica en Red de Cáncer (CIBERONC), Av. Monforte de Lemos, 3-5. Pabellón 11, Planta 0, 28029 Madrid, Spain

**Keywords:** endometrial cancer, liquid biopsy, preclinical models, organoids, uterine aspirate, cfDNA, MSI, monitoring, immunotherapy, precision oncology

## Abstract

Endometrial cancer (EC) is the 4th most common neoplasm of the female genital tract, with 15–20% of patients being of high risk of recurrence which leads to a significant decrease in patient survival. Current therapeutic options for patients with EC are poor, being the combined therapy of carboplatin and paclitaxel the standard of care, with limited efficacy. Therefore, new therapeutic options and better monitoring tools are needed to improve the management of the disease. In the current case report, we showcase the value of liquid biopsy analyses in a microsatellite instability EC patient with initially good prognosis that however underwent rapid progression disease within 6 months post-surgery; through the study of plasma cfDNA/ctDNA dynamics to assess the tumour evolution during treatment, as well as the study of the uterine aspirate as a valuable sample that captures the intra-tumour heterogeneity that allows a comprehensive genomic profiling of the disease to identify potential therapeutic options. Furthermore, preclinical models were generated at the time of tumour progression to assess the efficacy of the identified targeted therapies.

## 1. Introduction

Endometrial cancer (EC) is the 4th most common neoplasm for females with its incidence has been increasing over the past decade [1]; however, it is usually associated with a good prognosis being around 4% of the total deaths due to this malignancy [2]. Nevertheless, 15–20% of endometrial tumours exhibit an aggressive phenotype, with a high risk of recurrence, and the survival rate within 5 years decreases down to 10% [3]. Nonetheless, regardless of recent updates on the clinical management of other tumour types directed to more personalised approaches, the management of EC patients has not really changed in the past years and more precise follow-up tools could be of great use for the management of the disease. In this regard, liquid biopsy has emerged as a successful tool to monitor the disease evolution in real time through the analysis of circulating biomarkers such as circulating tumour DNA (ctDNA) or circulating tumour cells (CTCs) [4]. Some studies have proven the clinical value of ctDNA analysis in longitudinal samples for disease monitoring in various cancer types, including EC [5,6,7,8,9,10,11], not only to predict prognosis [12,13] but also to assess therapy efficiency. Moreover, in the context of EC, the analysis of the uterine aspirate (UA) has proven to be a reliable sample to assess the mutational background of the disease, being able to recapitulate the intratumor heterogeneity [6,14].

Traditionally, personalised medicine has focused on the generation of in-vitro and in-vivo models to study the disease with the main limitation being the lack of heterogeneity and more complex physiological characteristics of the human body. Patient-derived organoid (PDOs) preclinical models have arisen as a powerful tool for personalised medicine that bypasses said limitations since they are a more realistic heterogenous culture that can recapitulate the tissue of origin and allow for drug screening assays as well as to identify the most effective treatment in each patient according to their mutational profile [15,16]. Furthermore, these PDOs have been successfully established from explants and tissue biopsies in patients suffering from gynaecological malignancies with a moderate success rate that allowed for the identification of potential therapies and the subsequent cell viability assays [17,18,19,20,21,22].

In the present case report, we apply different liquid biopsy analyses to monitor the disease evolution and to identify the mutational landscape for potentially targetable pathways in a microsatellite instability (MSI) EC patient with an initial clinical good prognosis that nevertheless underwent rapid progression disease within 6 months after surgery and was subjected to several treatments without clinical benefit; therefore, preclinical models were generated from the tumour metastasis for drug screening and therapy selection.

## 2. Results

### 2.1. Clinical Course

In March 2021, the patient was diagnosed with a G1 endometrioid adenocarcinoma of the endometrium, apparently stage I clinically and by imaging (MRI), although with two nonspecific nodes <10 mm. A total abdominal hysterectomy, bilateral adnexectomy and lymphadenectomy were performed. A definitive pathology report showed mixed areas of a low-grade and a high-grade adenocarcinoma of the endometrium, infiltrating less than 50% of the myometrium, without vascular, perineural or cervical stroma invasion; pelvic nodes were free of disease (0/29) but it infiltrated paraaortic nodes (1/4). Immunohistochemistry analysis showed deficient expression of MSH2 and MSH6 proteins, positive expression of the ER receptor (100%) and PR receptors (30%) and negative expression of HER2. Based on the above finding, she was diagnosed with a mixed tumour with low-grade and high-grade areas of an endometrioid adenocarcinoma of the endometrium, FIGO stage IIIC2, pT3bN0M0.

However, before starting the plan defined in the Multidisciplinary Committee (adjuvant chemotherapy (CTP) and radiotherapy (RTP)) based on the available clinical information, the patient went to the Accident and Emergency department (April 2021) due to the appearance of bleeding, associated with a lesion on the anterior side of the vagina, infiltrating the urethra. The lesion was biopsied and confirmed to be of endometrial carcinoma origin. Therefore, in line with these findings, the plan was modified, and it was decided to start CTP as soon as possible and, if feasible, later local therapy.

Between 2 May 2021 and 1 July 2021, she received 3 cycles of carboplatin-paclitaxel, well tolerated with a PS-ECOG = 0 but evaluated as progression by image. Between 2 August 2021 and 6 September 2021, she received concomitant cisplatin at 40 mg/m^2^ and RTP and determined again as clinical and imaging locoregional progression with the appearance of an implant in the abdominal wall also confirmed by biopsy to be of endometrial carcinoma origin, that was also used for the generation of preclinical models (Figure 1A). The patient had a clinical impairment, with significant pelvic pain and a PS-ECOG = 2. Between 20 October 2021 and 3 December 2021, she received 3 cycles of Pembrolizumab, an anti-PD-1 antibody treatment approved by the FDA for the treatment of advanced metastatic EC patients with MSI-H or dMMR that has shown an increase in the median overall survival [23]. Nevertheless, the patient underwent progressive disease, assessed also in the physical examination (more abdominal implants), with pain and diarrhoea as the main symptoms, causing a progressive and definitive clinical impairment to PS-ECOG = 3 (Figure 1A,B). Despite the efforts, the patient died on 6 January 2022.

### 2.2. Liquid Biopsy Analysis as a Powerful Monitoring Tool to Identify Recurrences and Monitor Treatment Efficiency

As to confirm the molecular subtype of the patient, multiple analyses were performed in the UA, a reliable minimally invasive sample that captures the intratumor heterogeneity [14]. First, to identify the mutational landscape, the UA was subjected to next generation sequencing (NGS) analysis using the Oncomine Panel V3; alterations were found in *ARID1A, MSH2, NFE2L2, PIK3R1A, NBN, PTEN* and *TP53* (Figure 2A). Based on the alterations, the patient was classified as MSI-H due to the alterations found in *MSH2* [24]; nevertheless, alterations in *TP53* and *NBN* are indicative of a poor prognosis of the patient, since alterations on *NBN* have been linked to cisplatin resistance in some cancer subtypes [25,26] and *TP53* is strongly associated with shorter progression-free survival and overall survival in EC [24,27]. Moreover, the mutational landscape was later used to identify potential active therapies based on the found alterations. 

Due to the ambiguous molecular classification of the patient, the UA was further characterised by analysing the status of 5 mononucleotide repeats (BAT25, BAT26, NR24, NR21 and Mono27) using a ddPCR-MSI panel. Following this approach, over 20% of variant allelic frequency (VAF) was found in all 5 mononucleotide repeats; therefore, the patient was classified as MSI-H.

After the tumour mutational landscape was analysed in the UA, we assessed the plasma cfDNA/ctDNA status in the patient as a monitoring tool. The cfDNA concentration was assessed using Qubit fluorometry. High levels of cfDNA were found at baseline (>30,000 GE/mL) but decreased after treatment (<3000 GE/mL), showcasing the impact of debulking surgery. Importantly, during tumour progression, cfDNA levels were representative of the tumour kinetics, increasing over time, reflective of the lack of effect during RTP and systemic treatments (Figure 1B).

Since the patient presented deficiency in the MMR proteins and was confirmed to have all 5 MSI markers altered in the UA, the same ddPCR-MSI panel was used to analyse the ctDNA dynamics. With this approach, we could confirm that ctDNA dynamics mirror the disease evolution (Figure 1B). Thus, at surgery, high levels of alteration were found in all 5 MSI markers; while, after surgery, the ctDNA levels decreased, evidencing the efficiency of debulking surgery. As the patient was undergoing such rapid progression, ctDNA levels were closely monitored after the progression was detected, and as seen in Figure 1, the levels become more prominent at disease progression, indicative of the increase in tumour burden. During pembrolizumab treatment, ctDNA levels kept on increasing reflecting the lack of efficacy of the treatment.

### 2.3. Preclinical Models for the Identification of Efficient Therapies

At the time of progression disease, PDOs were generated from the progression tissue biopsy by mechanically and enzymatic dissociation of the tissue and cultured with conditioned media that promotes cell proliferation and survival. In order to ensure that the generated PDOs accurately resemble the disease, they were characterised by IHC and NGS analysis and compared to the metastatic lesion. As for the molecular profile, the generated PDOs shared the same IHC expression as the peritoneal metastasis, with aberrant expression in the 4 MMR proteins, negative expression of the receptor tyrosine-protein kinase erbB-2 (HER2), oestrogen receptors (ER) and progesterone receptors (PRA) negative and aberrant expression of TP53 (Figure 2B). As for the genetic landscape, the mutational profile of the PDOs was compared to the UA. Both samples shared the same mutations and MSI profile, but with higher percentages of alteration in the generated PDOs; only one discrepancy in the *PIK3R1A* gene was found, being the alteration found in a different codon; nevertheless, with the same clinical impact: a non-functional protein (Figure 2A).

After confirming that the PDOs resemble the tissue of origin, they were subjected to alternative therapies, based on the mutational profile the following compounds were selected: carboplatin-paclitaxel combined therapy, the standard of care, olaparib, a Poly (ADP-ribose) polymerase inhibitor involved in DNA repair pathways; trabectedin, an antineoplastic compound that has shown high efficiency on the treatment of advance ovarian and endometrial carcinomas harbouring *ARID1A* mutations [28]; and gemcitabine, another potent antineoplastic compound involved in DNA replication.

Cell viability was assessed in PDOs using AB after 72 h from the beginning of the assay. A statistically significant reduction in cell viability was found when PDOs were treated with gemcitabine (two-way ANOVA; *p*-value > 0.01) in contrast to the remaining therapies where no significant reduction in the cell viability was found (Figure 2C,D) just as in the patient, where tumour progression happened while the patient was being subjected to CTP and RTP treatment. These results suggest the efficacy of gemcitabine as a potential treatment; nevertheless, due to the clinical impairment, it could not be administered to the patient.

## 3. Discussion

The implementation of the molecular characterisation in EC, which further stratified patients based on the genetic profile [24] and its implementation into the clinical management of the risk of recurrence [29], allowed to limit the under or overtreatment of EC patients. Nevertheless, some molecular subtypes still lack a clear clinical outcome [27] and the implementation of further molecular stratification such as the *CTNNB1* mutant [27,30] could help identify patients with poor prognosis and improve their clinical management, or as in the present case report, where there is ambiguity regarding the molecular subtype where the patient should be included.

Moreover, there is a clear need for more precise follow-up tools that would benefit EC management to predict early recurrences, disease progression and even response to therapy. In this regard, liquid biopsy studies have proven their value as they are easily obtainable samples that provide information in real time, and improve the understanding of tumour heterogeneity [31]. In the context of EC, recent studies have shown the value of circulating extracellular vesicles (cEVs) present in the blood, ascites, or urine as potential biomarkers in the clinical management of gynaecological tumours [32]; for instance, Annexin A2 and L1 cell adhesion molecule have shown promising value as biomarkers for early detection and prognosis in endometrial tumours [33]. Metabolomics has also arisen as a valuable approach to predicting tumour behaviour and pathological characteristics in endometrial tumours [34]. Another valuable minimally invasive sample, the UA, has proven to recapitulate the mutational profile found in tissue samples from ECs and is a representative sample of the molecular heterogeneity of primary carcinomas [6,14]. With this approach, we were able to characterise the genetic profile of the patient, identifying alterations in *ARID1A*, *MSH2*, *NFE2L2*, *PIK3R1A*, *NBN*, *PTEN* and *TP53* as well as instability in five mononucleotide repeats (BAT25, BAT26, NR24, NR21 and Mono27). The mutational landscape of the patient was indicative of poor prognosis due to the alteration found in *TP53*, linked to shorter overall survival in EC [24,27] as well as the presence of alterations in *NBN*, which has been previously reported to be involved in cisplatin resistance [25,26]; moreover, the mutational landscape was later used to identify potential active therapies based on the found alterations.

In addition to the UA analysis, cfDNA/ctDNA levels have been studied as a potential biomarker in EC [7,8,11] finding higher levels in patients than in healthy controls. Moreover, more aggressive phenotypes of the disease have been associated with higher cfDNA levels [6,8]. Furthermore, ctDNA analysis has also been performed in EC patients, determining its presence by means of NGS analysis [5] or ddPCR [6,12,13,35] finding the presence of ctDNA in around 30–40% of patients with localised disease. Moreover, Pereira et al. as well as Feng et al. showcased the clinical value in progression-free survival and overall survival of ctDNA in a retrospective cohort of gynaecological tumours (ovarian and endometrial tumours) with detectable levels of ctDNA [12,13]. In the present case report, longitudinal analysis of cfDNA/ctDNA samples proves the combined value of both biomarkers to follow-up the disease, as they mirror the tumour kinetics during chemo, radio and immunotherapy providing information in real time in a non-invasive manner. The implementation of cfDNA and ctDNA analysis to monitor the disease in the clinic could provide important insights into tumour behaviour in response to treatment and as a potential indicator of progressive disease that could help guide clinicians in therapy selection; therefore, translating into a more personalised medicine and improving patient care. Especially in cases such as the one shown in the current case report, where a patient with an initial clinical good prognosis ended up undergoing rapid progression disease within 6 months after surgery and was subjected to several treatments without clinical benefit.

Therefore, multidisciplinary approaches that combine multiple non-invasive predictors such as plasma ctDNA mutation and methylation profiling, cEVs, proteomic or metabolomics could be the key for a more personalised medicine and stratifying patients using minimally invasive procedures and help clinicians identify those patients with really poor clinical outcome and avoid under or overtreatment of the disease.

Despite recent improvements, there is a clear need for more effective treatments after chemotherapy resistances arise in patients with recurrent disease. In this context, PDOs have arisen as a key element in personalised medicine, as they are a heterogeneous sample that recapitulates the tissue of origin and can be used to perform high-throughput pharmacological studies in order to identify alternative therapeutic options [15,20] and, in the case of EC, they have been successfully established from tumour biopsies and tissue explants [18,21,22]. Our results prove the value of PDOs generation for personalised treatments, especially the extra value of using the tissue obtained from the PD biopsy. Therefore, more representative of the actual disease, as seen by the IHC and genomic profiling of the disease showing the same profile between the patient’s disease and the generated model, which translates into more trustworthy results and allows the identification of a potential therapy in gemcitabine after the pharmacological screening.

## 4. Materials and Methods

### 4.1. Patient Inclusion

The study was carried out according to the rules of the Declaration of Helsinki of 1975, revised in 2013, and according to the standards for good clinical practice and other local ethical laws and regulations. Informed consent form, approved by the pertinent ethical committees, was signed by the patient (Autonomic Galician Ethical Committee Code 2017/430).

### 4.2. Sample Collection and Processing

Plasma samples were processed as previously described [6]. In brief, a two-step centrifugation was performed to isolate the plasma samples. CfDNA was isolated with the QIAamp Circulating Nucleic Acid Kit (QIAGEN, Hilden, Germany) and quantified using Qubit (ThermoFisher, Waltham, MA, USA). CtDNA was assessed by droplet digital PCR (Bio-Rad Laboratories, Hercules, CA, USA), using a specific ddPCR assay to determine the status of 5 mononucleotide repeats and run on a QX-200 dPCR system (Bio-Rad Laboratories, Hercules, CA, USA) using TaqMan chemistry according to the specific MSI profile of the patient.

The UA was obtained at surgery with a Cornier cannula and kept on ice until they were processed as previously described [6]. In brief, the UA was homogenised with phosphate-buffered saline (PBS) at a 1:1 ratio and centrifuged at 4 °C for 20 min at 2500× *g*.

Solid biopsy was obtained during re-biopsy of the progression disease and immediately processed for the generation of patient-derived organoids as described below.

### 4.3. Preclinical Model Generation

Patient-derived organoids (PDOs) were generated from the progression solid biopsy as previously described [18]. In brief, tissue was mechanically and chemically dissociated by mincing the tissue and incubation with DMEM-F12 (Lonza, Basel, Switzerland) with 1.25 U/mL of Dispase II (Sigma-Aldrich, St. Louis, MO, USA) and 0.4 mg/mL of Collagenase IV (ThermoFisher, Waltham, MA, USA) during 2 h at 37 °C; next, cell chunks were dissociated into a single cell by incubation at 37 °C with TrypLe (ThermoFisher, Waltham, MA, USA). Cells were seeded in basement membrane extract (BME) (R&D Systems, Minneapolis, MN, USA) domes with conditioned medium as already described in the literature [18] (Table 1).

### 4.4. Drug Screening

For drug screening assays, cells from the generated PDOs were dissociated into single cells and seeded in a p96-well plate in BME domes with depleted medium, after 48 h medium was supplemented with the specific treatment: Carboplatin-paclitaxel combined therapy at a 2:1 ratio; doxorubicin; trabectedin; gemcitabine; olaparib or vehicle. Viability was assessed using the alamarBlueTM (AB) approach at 72 h (ThermoFisher, Waltham, MA, United States) according to the manufacturer’s indication.

### 4.5. Immunohistochemistry Characterisation

For IHC characterisation, organoids were removed from BME domes, fixed with 4% PFA (Ted Pella, Inc., Redding, CA, USA) for 24 h at 4 °C, and then embedded into 1% agar tissue-teks and stored in 70% ethanol until processed as previously described [36].

The immunohistochemistry (IHC) characterization was performed at the Pathology Services from the hospital. Sections from FFPE tumour tissue (4 μm thick) were automatically stained in a Dako Omnis immunostainer (Dako-Agilent, Santa Clara, CA, USA). Briefly, the immunohistochemical protocol included the following steps: (1) heat-induced epitope retrieval solution at high pH (Dako-Agilent) for 20 min at 97 °C; (2) ready-to-use FLEX primary monoclonal antibodies (Dako-Agilent); (3) mouse and rabbit linker (Dako-Agilent) for 10 min each; (4) EnVision FLEX/HRP (Dako-Agilent) for 20 min; (5) 3,3′-diaminobenzidine tetrahydrochloride chromogen solution (Dako-Agilent) for 5 min; and (6) EnVision FLEX hematoxylin (Dako-Agilent) for 5 min. Adjacent normal glands and surrounding lymphocytes were employed as internal positive controls.

### 4.6. Sequencing Analysis

DNA from the UA and generated PDOs was sequenced using the Oncomine V3 Panel (ThermoFisher, Waltham, MA, USA) as previously described [6]. In brief, to prepare amplicon libraries, we performed targeted sequencing of the samples with the multiplex PCR with the Ion AmpliSeq Library Kit 2.0 and Oncomine Comprehensive Panel v3 (ThermoFisher, Pleasanton, CA, USA). For PCR, a total of 17 and 20 cycles were performed. The PCR template preparation and enrichment were performed with the Ion PGM Template OT2 200 Kit and Ion OneTouch 2 System. Finally, the Ion PGM Sequencing 200 Kit v2 and Ion PGM System (Life Technologies, Santa Clara, CA, USA) were used for DNA sequencing, according to the manufacturer’s protocols. Duplicates were analysed for 10% of the samples and found to yield equivalent results.

For the bioinformatics analysis, alignment to the Hg19 human reference genome and variant calling were performed with Torrent Suite™ Software v.4.2.1 (Life Technologies, Santa Clara, CA, USA). All samples were sequenced and analysed in comparable conditions. The mean coverage per sequenced sample was approximately 1500 reads per base. Variants with a Phred quality score field value less than 100 were considered low-quality variants. The prediction of genomic variant effects on protein function was performed with the PROVEAN Genome Variants tool (http://provean.jcvi.org/index.php). Variants with possibly damaging or deleterious consequences, as predicted by at least one of the PROVEAN predictors, were considered to be of interest and were visually checked with Integrative Genomics Viewer (IGV) v.2.3.40, Broad Institute. Variants with a global minor allele frequency above 0.05% were considered single nucleotide polymorphisms and were rejected (data from dbSNP, http://www.ncbi.nlm.nih.gov/SNP/).

## 5. Conclusions

In summary, our approach showcases the value of combining the analysis of the UA, to characterise the mutational profile of the patient; follow-up of the disease using the plasma biomarkers ctDNA to understand the disease evolution and the generation of PDOs to identify potential therapies that could be implemented into the clinic. Overall, this combination would improve the management of EC patients opening new avenues for a more personalised precision medicine in the field of gynaecological cancers.

## Figures and Tables

**Figure 1 ijms-23-08539-f001:**
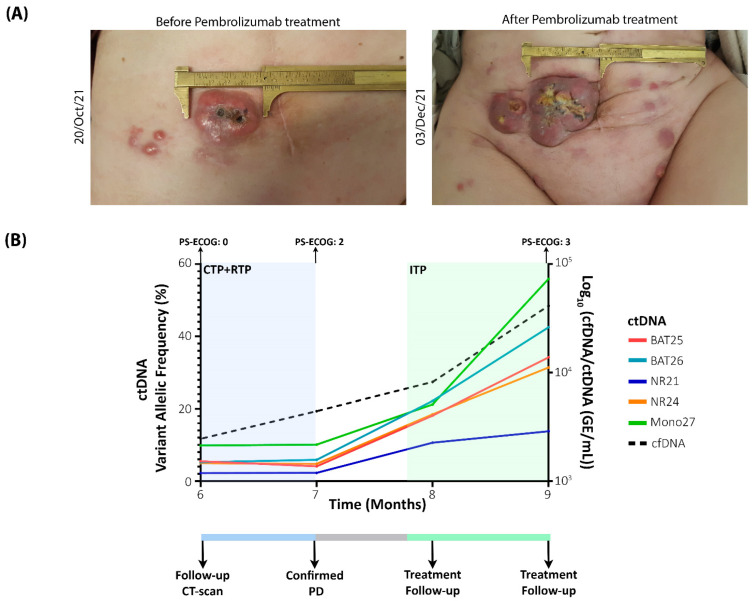
(**A**) Representative images of the abdominal implants showing the rapid tumour progression during pembrolizumab treatment. (**B**) Top panel corresponds to the graphical representation of the disease evolution showcasing the value of cfDNA/ctDNA analysis to monitor response to therapy, the kinetics of both biomarkers mimic the tumour evolution with their levels rapidly increasing after the CT scan and during treatment with CTP and RTP (blue background) and shortly after the confirmation of progression disease levels keep increasing, showing no response to ITP (green background). Bottom panel corresponds to the different events where longitudinal samples were collected. CTP: chemotherapy. RTP: radiotherapy. ITP: immunotherapy. PD: progression disease.

**Figure 2 ijms-23-08539-f002:**
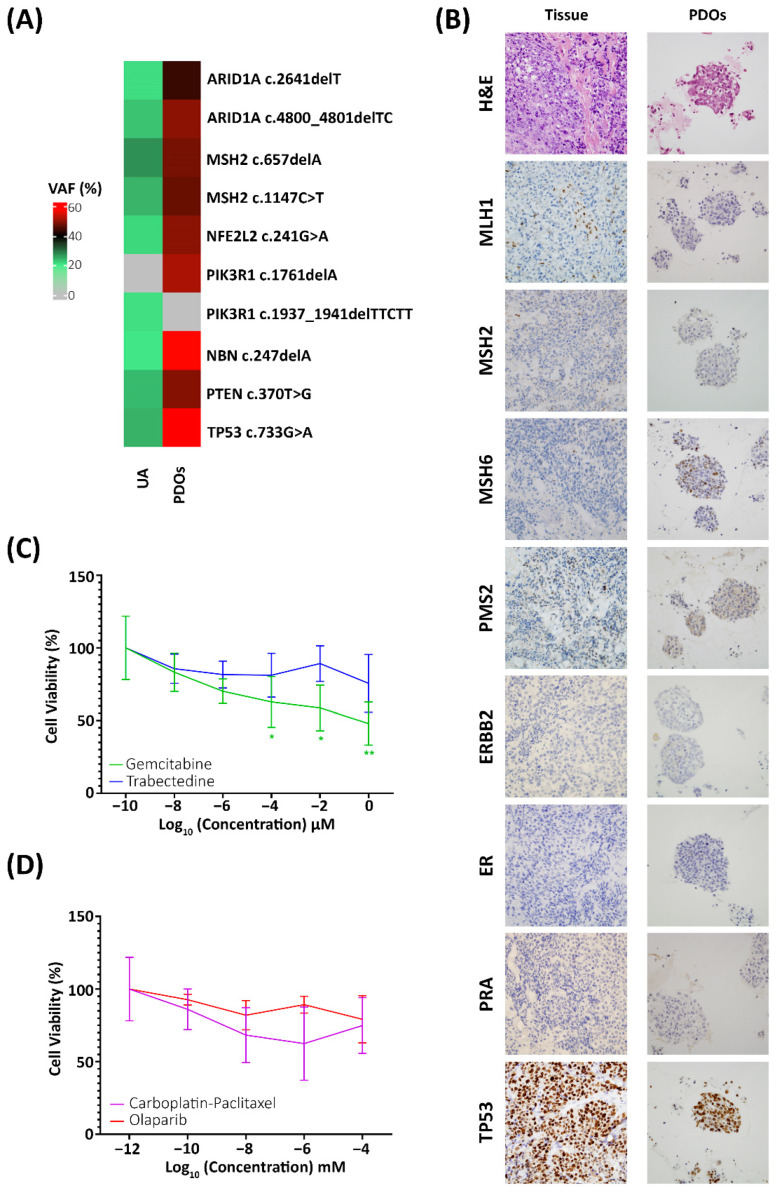
(**A**) Heatmap representation of the mutational profile of the UA and the generated PDOs showing the same mutational profile. (**B**) Immunohistochemistry comparison between the tissue biopsy and the generated PDOs reflecting the same molecular profile. (**C**) Cell viability analysis after the treatment with gemcitabine (green) and trabectedin (blue) showing a significant decrease in cell viability when treated with gemcitabine. (**D**) Cell viability analysis after the treatment with carboplatin-paclitaxel combined therapy (purple) and Olaparib (red) showing no significant reduction in cell viability. Two-way ANOVA * *p*-value < 0.05, ** *p*-value > 0.01.

**Table 1 ijms-23-08539-t001:** List of compounds, concentration and supplier for the organoid culture medium.

Product	Concentration	Supplier	Catalog Number
DMEM/F12	-	Lonza	12-719F
h-R-spondin1 (hRSPO1)	7.50 nM	Peprotech	120-38
h-noggin (hNOG)	2.17 nM	Peprotech	120-10C
B27 supplement	2%	ThermoFisher	17504044
N2 supplement	1%	ThermoFisher	17502048
Insulin-transferrin-selenium (ITS)	1%	ThermoFisher	41400045
GlutaMAXTM supplement	1%	ThermoFisher	35050061
Antibiotic-antimycotic	1%	ThermoFisher	15240062
Nicotinamide (NICO)	2 mM	Sigma-Aldrich	N0636
A83-01	0.6 µM	Sigma-Aldrich	SML0788
N-acetyl L-cysteine (NAC)	1.25 mM	Sigma-Aldrich	A7250
EGF	8 nM	Peprotech	AF-100-15
b-FGF	0.1 nM	Peprotech	100-18B
SB202190 (p38i)	10 µM	Sigma-Aldrich	S7067
17-β estradiol	1 nM	Sigma-Aldrich	E8875
Y-27632 *	10 µM	Selleckchem	S1049

* Y-27632 only for organoid formation or dissociation at passaging.

## Data Availability

The datasets used and/or analysed during the current study are available from the corresponding author on reasonable request.

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
