# Peer review of "Improving the Management of Endometrial Cancer Patients through the Use of Liquid Biopsy Analyses: A Case Report"

_ijms, 2022, doi:10.3390/ijms23158539_

Round 1

Reviewer 1 Report

Casas-Arozamena et al., in their case report "Improving the Management of Endometrial Cancer Patients Through the Use of Liquid Biopsy Analyses: A Case Report," describes the use of various liquid biopsy analyses to monitor disease evolution and identify the mutational landscape for potentially targetable pathways in microsatellite instability (MSI) EC patient with an initial clinical good prognosis who nonetheless experienced rapid disease progression within 6 months. It is an intriguing study that describes the clinical management of endometrial cancer. Their study includes the following questions:

1. The authors employed an NGS panel to show changes in several genes, although their significance in EC progression is unknown.

2. The wording lacks a rationale for testing MSI.

3. What was the antibody dilution used in the IHC study?

4. Dic should be replaced with Dec in Fig 1A.

5. The representation of Fig 1B is perplexing and difficult for readers to comprehend.

Reviewer 2 Report

In this case report, Authors described the capacity of uterine aspirate UA to be a minimally invasive and a representative sample of the mutational profile of primary endometrial carcinoma (EC). In addition, Authors analyzed cfDNA and ctDNA to monitor the disease after primary treatment and found that these markers represent potential indicators of progression of disease that could help to guide clinicians in follow-up and in therapy selection. 

The approach described in this case report represents an innovative idea, since, novel molecular prognostic factors have been demonstrated to be accurate in the prediction of the prognosis of EC, as reported in the ESGO/ESTRO/ESP guidelines for the management of EC, but such molecular assessment has been applicated only on endometrial biopsies or surgical specimens, not in circulating tumor cells or DNA.

I have the following comments to the Authors:

·      Please be consistent about the epidemiology of EC throughout the manuscript, since abstract and introduction report different information.

·      Line 81: “bilateral” instead of “double” would sound more appropriate.

·      Line 100: Please specify what the abbreviation “CDDP” stands for.

·      Molecular prognostic factors have been demonstrated to be accurate in the prediction of the prognosis of EC, as reported in the ESGO/ESTRO/ESP guidelines for the management of EC. As Authors wrote, this approach is able to further stratify patients based on the genetic profile, but there is a clear need for more precise follow-up tools. Authors should expand discussion describing other potential clinical implications of the application of molecular characterization in the management of EC (e.g. PMID: 34073635).

·      Early and minimally invasive diagnosis and risk assessment of EC is a trending topic in literature. The ESTRO/ESGO/ESP guidelines for the management of EC proposed a novel risk stratification model including TCGA molecular groups (PMID: 35078650) to assess the prognosis of EC. In addition to molecular factors for the prognosis assessment, metabolomics has recently appeared as a promising test for a non-invasive diagnosis of several diseases and metabolites were found able to predict the presence of EC (PMID: 32180221), tumor behavior (progression and recurrence) and pathological characteristics (histotype, myometrial invasion and lymph vascular space invasion). Is there any evidence to apply metabolic predictors or any other non-invasive predictor apart from ctDNA of cfDNA? If so, could these novel predictors be integrated with each other in a multimodal diagnostic model? This could have an extraordinary impact on the management of EC and Authors should discuss about this. 
